# Macrophage Repolarization as a Therapeutic Strategy for Osteosarcoma

**DOI:** 10.3390/ijms24032858

**Published:** 2023-02-02

**Authors:** Namrata Anand, Keng Hee Peh, Jill M. Kolesar

**Affiliations:** 1Department of Pharmacy Practice and Science, College of Pharmacy, University of Kentucky, Lexington, KY 40508, USA; 2Markey Cancer Center, University of Kentucky, Lexington, KY 40508, USA

**Keywords:** M1 macrophage, M2 macrophage, osteosarcoma, TAM, polarization, TME, repolarization

## Abstract

Macrophages are versatile immune cells and can adapt to both external stimuli and their surrounding environment. Macrophages are categorized into two major categories; M1 macrophages release pro-inflammatory cytokines and produce protective responses that lead to antimicrobial or antitumor activity. M2 or tumor-associated macrophages (TAM) release anti-inflammatory cytokines that support tumor growth, invasion capacity, and metastatic potential. Since macrophages can be re-polarized from an M2 to an M1 phenotype with a variety of strategies, this has emerged as an innovative anti-cancer approach. Osteosarcoma (OS) is a kind of bone cancer and consists of a complex niche, and immunotherapy is not very effective. Therefore, immediate attention to new strategies is required. We incorporated the recent studies that have used M2-M1 repolarization strategies in the aspect of treating OS cancer.

## 1. Introduction

Osteosarcoma (OS) is a type of rare bone sarcoma with an estimated 3910 new cases in 2022. OS originates in mesenchymal stem cells that produce osteoblasts, which are the cells responsible for new bone. Uncontrolled proliferation can lead to the formation of osteoid; a type of unmineralized/immature bone that can cause a primary bone tumor [1]. The onset is bimodal, occurring most commonly in children and adolescents and adults older than 65. The most common sites are the long bones of the legs and the knee area at the joint, but it rarely occurs in the pelvis, spine, or visceral organs [2,3]. Surgery is the primary therapy for OS, and systemic chemotherapy, either in the adjuvant or neoadjuvant setting, is also routinely administered. While there is no consensus on the optimal timing or regimen, many centers prefer neoadjuvant chemotherapy, and the most common regimen is MAP (methotrexate, doxorubicin, and cisplatin). Radiotherapy is typically limited to patients that are ineligible for surgery, that have unresectable tumors, or that have the presence of residual disease post-surgery. Chemotherapy in the adjuvant or neoadjuvant setting has improved the 5-year survival rate from 20% to 65% primarily by eliminating metastasis [4]. Despite these advances, 35% of patients still die from their disease and novel treatments are urgently needed [5].

Immune checkpoint inhibitors targeting Programmed death-ligand 1 (PD-L1), Programmed death-1 (PD-1), and CTLA-4 have been approved for several cancer types and have resulted in meaningful improvements in overall survival. However, despite OS cells expressing PD-L1, clinical trials of immune checkpoint inhibitors in OS to date have been disappointing [6,7,8]. The purpose of this review was to evaluate the immune status of OS and identify alternative strategies for immune activation and anticancer activity beyond immune checkpoint inhibitors.

## 2. Osteosarcoma Tumor-Microenvironment

The tumor micro-environment (TME) in OS is a very complex niche comprised of bone cells, immune cells, vascular cells, and endothelial cells such as mesenchymal stem cells (MSCs). MSC cells are mainly derived from bone marrow and non-hematopoietic precursor cells, and they bear the tendency to differentiate into chondrocytes and adipocytes and many extra-mesodermal cell types [9]. Mutations in p53 and Rb pathways, which are key regulators of MSC development, are associated with the development of OS [10,11]. P53 deletion has been linked with increased osteoblast differentiation, thereby impairing the osteocyte terminal maturation and leading to immature bone cells [12], whereas Rb deletion/mutation has been associated with the differentiation of mesenchymal cells and osteogenic and adipogenic lineage through various transcriptional pathways [13,14]. Rat experimental model-based studies have suggested that MSCs present in the bone microenvironment can enhance tumor progression and pulmonary metastasis. These studies have suggested that the undifferentiation of bone marrow-derived MSCs could lead to cells responsible for OS development under the signal provided by the TME [15].

Another component of bone-OS niche includes bone-forming cells (e.g., osteoblasts, osteocytes, and osteoclasts) that are responsible for bone tissue formation and the maintenance of the homeostatic environment inside the bone. Osteoblasts suppress the activity of osteoclasts, facilitating new bone formation. These cells are responsible for preserving the interaction between the bone matrix and immune cells for normal proliferation and maintenance via the secretion of various chemokines, cytokines, and signaling molecules [16,17,18]. Genetic alterations such as DNA methylation and histone modifications are also associated with changes in osteoblast differentiation and function, leading to the development of OS [19,20].

The primary tumor also contains tumor-associated macrophages (TAM) that promote a “pro-tumor” environment by producing anti-inflammatory cytokines such as IL-10 and TGF-beta via the stimulation of IL-13 and IL-4. TAMs consist of a mixed population, including long-lived macrophages derived from the yolk sac and short-lived macrophages, originating from bone marrow and circulating monocytes [21]. The presence of TAMs allows OS cells to escape the immune response and develop in a tumor-promoting niche [22]. Other immune cells present in the TME are comprised of classical M1 macrophages, T cells, B cells, lymphocytes, effector molecules, and natural killer cells (NKs) which demonstrate anti-cancer properties [23]. The overabundance of the M2 phenotypic macrophages promotes pro-tumor and immunologically “cold” cancer, and therefore, the re-polarization of cancers from the M2 to M1 phenotype is a novel anti-cancer strategy [24].

## 3. Macrophages Involved in Bone Microenvironment

There are three distinct populations of macrophages that are known to be present in the bone TME: bone marrow macrophages, osteoclasts, and osteomacs, which are a special type of macrophage residing in bone tissue overlying mature osteoblasts [25,26]. The generally defined role of osteomacs has been associated with the role of immunosurveillance in the bone environment and was further confirmed to provide mineralization to the bone, which was previously thought to be a function of osteoblasts [25].

The macrophage population in the bone TME shows plasticity/polarization towards different immune stimulating environments. There are two major different types of macrophages found in the bone microenvironment, classical macrophages derived from circulating monocytes and alternative macrophages that are stimulated by the Th2-type of cytokines and accumulate at the tissue site. These are further sub-classified as M1 (classical macrophages), polarized by interferon-gamma (IFN-Ƴ) and lipopolysaccharide (LPS) and that promote Th1-type immune responses in the TME niche through their phagocytic activity and the production of pro-inflammatory cytokines, including tumor necrosis factor-alpha (TNF-α), reactive oxygen species (ROS), and inducible nitric oxide synthetase (iNOS). These activated/polarized macrophages secrete chemokines, CXCL9 and CXCL10, which help to control proliferating tumor cells. M1 macrophages often show increased expression of major histocompatibility II and surface glycoproteins CD80/86 and, therefore, act as antigen-presenting cells (APCs) for T cell priming and exerting proinflammatory responses [27].

M2 macrophages (alternative macrophages) correspond to the TAM phenotype, are polarized by interleukin 4 (IL-4) and IL-13, and produce an anti-inflammatory response by secreting IL-10, Arg-1, CXCR1, and CXCR2 [28,29]. Previous studies have suggested that M2 macrophages released anti-inflammatory cytokine, chemokines, and signaling cascades, showing an increase in the prognosis of OS and the progression of lung metastasis [30,31]. The majority of the population of TME, almost 50%, is comprised of a macrophage population in both primary and secondary metastasis, which is being polarized or converted into M2-like tumor-promoting macrophages, resulting in the poor survival of patients [1,24,32] (Figure 1).

TAM or M2 phenotypic cells present in TME also express surface markers such as CD47 and PD-L1 that produce the anti-phagocytic or “don’t eat me” signal to escape the immune response [33]. M2 macrophages have been further subdivided into four subtypes M2a, M2b, M2c, and M2d [34,35]. M2a macrophages are induced by IL-4 and IL-13, they express CD206, and they release TGF-beta. The M2b subtype is induced upon stimulation with TLR and IL-1R antagonist and secretes high levels of CCL1 and CD258 [36]. M2c types are stimulated by IL-10 via the activation of the STAT3 transcription factor [37]. M2d macrophages are the most important TAM subtype of M2 macrophages, and they are stimulated by TLR ligands or IL-6 and secret high levels of TGF-beta and VEGF. This subtype is mostly found in progressive primary bone tumors as well as in metastatic sites, resulting in increased angiogenesis via the secretion of VEGF and matrix metallopeptidase (MMP9), which help to promote tumor growth and proliferation [38,39].

In addition to macrophages, tumor-infiltrating leukocytes (TIL), dendritic cells (DCs), and mast cells also constitute the population of the OS TME, which can act as APCs [40,41]. IL-1beta is a proinflammatory cytokine secreted by M2, MSCs, and osteoblasts, and it helps to maintain the progressive stem cell type of cancer cells [42]. DCs and CD3^+^ T lymphocytes are also a major population in all types of bone sarcoma patients, which corresponds to the TAM infiltration [40]. Another recent study highlighted that the differential analysis of immune-related genes using a bioinformatics approach can be used to stratify patients into high and low risk to predict unfavorable survival outcomes in OS. Patients with a low risk of unfavorable survival outcomes were found to have increased macrophages and CD8 T-cells [43]. A previous study by Dupkar et al. suggested that the presence of CD146^+^ vascular cells and CD163^+^ M2 macrophages in lung metastases may be involved in promoting the neoangiogenic process [44]. The role of IL-34 was also linked with macrophage differentiation in OS using in vitro and in vivo models, and it enhanced the prognosis of the disease [45]. Overall, the heterogenicity of the bone TME is very complex and is regulated by the dominant macrophage phenotype, which determines the fate of the tumor (Figure 1). Therefore, a therapy able to convert TAM or M2 pro-cancer macrophages to anti-cancer M1-type macrophages could have significant clinical activity in patients with OS.

The presence of various cell types in the TME with pro-cancer and anti-cancer effects makes it an attractive target for the conversion of pro-cancer cells, such as TAMs, to anti-cancer M1 phenotypes. Strategies include using immune checkpoint inhibitors, such as small molecules able to re-polarize macrophages, which is in various phases of development and is described below.

## 4. Checkpoint Inhibitors in Osteosarcoma

### 4.1. Checkpoint Inhibitor’s Mechanism of Action

Checkpoint inhibitors have emerged as effective anticancer agents and are currently used in the front-line setting in multiple solid tumors [46]. Two cells play key roles in checkpoint inhibitor activity: T cells, which express the PD-1 receptor, and cancer cells or APCs, which express PD-L1, a PD-1 receptor ligand (Figure 2). When the PD-1 receptor and the PD-L1 ligand interact, a signal-inhibiting T cell receptor (TCR)-mediated activation occurs, which results in a lack of T-cell activity and continued tumor progression [47]. Another type of checkpoint inhibitor targets cytotoxic T lymphocyte protein 4 (CTLA-4) [47,48]. CTLA4 and CD28 are costimulatory receptors present on T cells with similar properties that interact with the ligands CD80 and CD86 present on APCs. CTLA4 interacts with its ligand with a much higher affinity than CD28, which inhibits the normal T cell stimulation via CD28 and the CD80/86 interaction required for normal T cell stimulation and its further activation by APCs. The immune complex formed by CTLA4 and CD80/86 suppresses further T cell activation and inhibits normal T cell function [49]. Blocking these immune checkpoints has enhanced the cytotoxic effects of T cells in various solid and hematologic tumors [50].

### 4.2. The Rationale for Checkpoint Inhibitors in Osteosarcoma

Despite the reported overexpression of PD-L1 and PD-1 in in vitro OS models using various cell lines and tumor tissues, the correlation between PD-L1 status and predicting clinical outcomes in clinical and observational studies remains unclear [51,52]. A previous study demonstrated that PD-L1 expression in patients is associated with a poorer likelihood of going five years event free survival (EFS) [53]. PD-1 was also found to be highly expressed in CD4^+^ and CD8^+^ cells and was associated with an increased risk of OS progression and metastatic disease [54]. Similar findings were observed in a meta-analysis of 14 studies that included 868 patients with OS that had higher PD-L1 expression in tumor tissue, which was associated with an increased risk of metastatic disease and worse survival outcomes [55]. Therefore, assessing checkpoint inhibitors as a therapeutic strategy in OS appears to be a rational approach.

### 4.3. Checkpoint Inhibitors Clinical Trials in Osteosarcoma

Despite previously reported increased PD-L1 expression in OS, checkpoint inhibitors alone or in combination with other therapies have limited activity in OS (Table 1) (Figure 2). In a single-arm phase 2 trial in unresectable and relapsed osteosarcoma, of the 12 patients treated with pembrolizumab, none (0/12) experienced clinical benefit, defined as complete response (CR), partial response (PR), or stable disease (SD), at 18 weeks [7]. Tawbi and colleagues also evaluated single-agent pembrolizumab in a phase 2 study, enrolling patients with unresectable and recurrent sarcoma. Only 1 of 22 (5%) patients with osteosarcoma achieved an objective response, defined as CR and PR [6]. Twelve patients with osteosarcoma were enrolled in the iMATRIX trial, which was a phase 1/2 trial of single-agent atezolizumab for adults and children with previously treated solid tumors or lymphoma. Of the ten patients with osteosarcoma who were assessable for a response, none achieved a CR, PR, or SD. Of the six patients with osteosarcoma with a PD-L1 assessment, only one had >5% positivity of tumor cells [56]. None of these trials required a PD-L1 positivity for inclusion, and the ability of PD-L1 to predict the response in osteosarcoma remains unknown.

Combination immunotherapies have been successful in previous disease states, such as in non-small cell lung cancer, melanoma, and hepatocellular carcinoma; therefore, utilizing combination immunotherapies is justified as a rationale strategy to overcome the lack of efficacy from single-agent immunotherapy [57]. A phase 2 study assessing the efficacy of single-agent nivolumab or combination nivolumab and ipilimumab in metastatic, relapsed, refractory OS demonstrated an ORR of 8% (3/38) and 15% (6/41), respectively [58]. Despite modest efficacy and meeting the study’s predetermined endpoint in the combination arm, there was only one patient with OS enrolled in the trial, and no efficacy data were reported. Another phase 2 study that enrolled 57 patients, assessing the efficacy of a combination of durvalumab and tremelimumab, demonstrated a 12-week PFS rate of 49% (28/57) [59]. However, a subgroup analysis of five OS patients demonstrated a 12-week PFS rate of 20% (1/5). Exploratory analysis from the trial identified an increase in infiltrating immune lymphocytes in responders with alveolar sarcoma, but PD-L1 expression at baseline did not correlate significantly with the response to the durvalumab and tremelimumab combination, suggesting that other immune biomarkers may be predictors of response to combination immune checkpoint inhibitors. A novel strategy combining nivolumab and an IL-2 agonist, bempegaldesleukin, was evaluated in a pilot study that enrolled 84 patients with relapsed, refractory, advanced, or metastatic sarcoma [60]. Of the 84 patients, 10 OS patients were enrolled, and nivolumab and bempegaldesleukin demonstrated an ORR of 0% (0/10). The exploratory genomics analysis in the pilot study demonstrated that, overall, patient tumors had a low tumor mutation burden. However, the subgroup analysis of sarcoma subtypes showed that OS had a statistically significant higher tumor mutation burden compared to other sarcomas. Despite OS patients having a statistically significant higher tumor mutation burden, there was no correlation between response and higher tumor mutation burden. Other cancers have demonstrated associations between high PD-L1 expression or tumor mutation and increased response to checkpoint inhibitors [60]. However, in OS, both biomarkers have not replicated similar findings. Previous combination immunotherapy trials only had a small number of OS patients enrolled; therefore, larger trials enrolling OS patients are needed. Other immune biomarkers aside from PD-L1 and tumor mutation burden must be explored to identify OS patients that will benefit from combination immunotherapy.

Checkpoint inhibitors have also been combined with other non-immune modulating agents in OS as a therapeutic strategy. Xie et al. evaluated camrelizumab in combination with a VEGFR-2 antagonist, apatinib, in a phase 2, single arm trial that enrolled 43 patients with locally advanced or metastatic relapsed refractory OS [8]. The combination therapy demonstrated a median PFS and overall survival of 6.2 and 11.3 months, respectively. The apatinib and camrelizumab combination demonstrated improved PFS and overall survival compared to the standard of care multi-kinase inhibitors, sorafenib and regorafenib, utilized in relapsed, refractory OS [8,61,62]. The inhibition of upstream signaling pathways was shown in in vitro models to downregulate PD-L1 expression and other immune biomarkers [63,64,65,66]. However, further research is needed to identify specific immune biomarkers associated with upstream signaling pathways to identify novel targets. Another phase 2 trial evaluated pembrolizumab in combination with an alkylating DNA damaging agent, cyclophosphamide, in 15 patients with relapsed refractory metastatic OS [67]. The combination therapy demonstrated a 6-month non-progression rate of 13.3% (2/15). Responders in the study had low PD-L1 expression, and there was no association with response.

**Table 1 ijms-24-02858-t001:** Summary of clinical trials assessing checkpoint inhibitors in osteosarcoma.

Drug/NCT Number/n (Osteosarcoma Subgroup)	Target Molecule	Trial Design	Inclusion Criteria	Primary Outcome	Time to Event Outcomes
Pembrolizumab NCT03013127 [7] *n* = 12	PD-1	Single arm, phase 2	≥18 y.o, r/r osteosarcoma	CBR: 0% (0/12)	Estimated mPFS: 1.4 m.o Estimated mOS: 6.6 m.o
Pembrolizumab NCT02301039 [6] *n* = 86 (*n*=22)	PD-1	Single-arm, phase 2	≥12 y.o, r/r locally advanced or metastatic sarcoma	ORR (bone sarcoma arm): 5% (2/40) ORR (osteosarcoma): 5% (1/22)	Bone sarcoma arm mPFS: 8 wk mOS: 52 wk
Atezolizumab NCT02541604 [56] *n* = 90 (*n* =12)	PD-L1	Single arm, phase 1–2	<30 y.o, r/r solid tumors and lymphomas	ORR (overall): 5% (4/87) ORR (osteosarcoma): 0% (0/10)	Overall mPFS: 1.3 m.o mOS: 7.4 m.o
Nivolumab + Ipilimumab NCT02500797 [58] *n* = 85 (*n* = 1) *	PD-1, CTLA-4	Non-comparative, randomized, two-arm, phase 2	≥18 y.o, r/r advanced or metastatic sarcoma	ORR (monotherapy): 8% (3/38) ORR (combination): 15% (6/41)	Monotherapy mPFS: 1.7 m.o mOS: 10.7 m.o Combination mPFS: 4.1 mo. mOS: 14.3 m.o
Durvalumab + Tremelimumab NCT02815995 [59] *n* = 57 (*n* = 5)	PD-L1, CTLA-4	Single arm, phase 2	≥18 y.o, r/r metastatic sarcomas	12 wk PFS rate (overall): 49% (28/57) 12 wk PFS rate (osteosarcoma): 20% (1/5)	Overall mPFS: 2.8 m.o mOS: 21.6 m.o Osteosarcoma mPFS: 1.81 m.o
Nivolumab + bempegaldesleukin NCT03282344 [60] *n* = 84 (*n* = 10)	PD-1, CD122 (IL-2)	Single arm, pilot study	≥12 y.o, r/r locally advanced or metastatic sarcoma	ORR (osteosarcoma): 0%	Osteosarcoma mPFS: 2.0 m.o mOS: 6.3 m.o
Apatinib + Camrelizumab NCT03359018 [8] *n* = 43	PD-1, VEGFR-2	Single arm, phase 2	≥11 y.o, r/r locally advanced or metastatic osteosarcoma	mPFS: 6.2 m.o	mOS: 11.3 m.o
Pembrolizumab + Cyclophosphamide NCT02406781 [67] *n* = 15	PD-1, alkylating agent (DNA damage)	Single arm, phase 2	≥18 y.o, r/r metastatic osteosarcoma	6 m.o non-progression rate: 13.3% (2/15)	mPFS: 1.4 m.o mOS: 5.6 m.o

CBR: clinical benefit rate, mPFS: median progression free survival, mOS: median overall survival, ORR: objective response rate, y.o: year old, m.o: months, wk: weeks, r/r: relapsed or refractory. * no reported outcomes in osteosarcoma subgroup.

Checkpoint inhibitors and combinations have been evaluated in several clinical trials and have demonstrated minimal clinical efficacy in OS. High PD-L1 expression in most studies have also consistently demonstrated a lack of association with immunotherapy response in OS. Other immune biomarkers and strategies targeting other immune cells outside of cytotoxic T-cells are needed in the treatment of OS.

## 5. Drugs Targeting Macrophage Repolarization in OS

Given the lack of activity of immune checkpoint inhibitors in OS, the repolarization of M2 macrophages to the M1 phenotype is a potential anti-cancer strategy, and a number of approaches are in development (Table 2).

### 5.1. ATRA

All-trans-retinoic acid (ATRA) is an active derivate of vitamin A with anti-tumor and anti-inflammatory properties and with roles in cellular differentiation, maintaining homeostasis and immune tolerance [75]. ATRA augments the NF-kB, ERK, and JNK pathways by inducing the secretion of IL-1 beta by human monocyte-derived macrophages, which causes a pro-inflammatory effect and the repolarization of M2 macrophages to M1 phenotype [76]. A study conducted by Zhou et al. demonstrated the antitumor and antimetastatic activity of ATRA through the inhibition of M2 polarization [68]. Bone marrow-derived macrophages were stimulated with M1 and M2 stimulators, but after treatment with ATRA, the expression of M2 surface molecules such as CD206 was found to be decreased. Additionally, the treatment halted the invasion and migration potency of OS cells when observed from in vitro and in vivo mice models [68]. These findings were further supported by another study and also suggested that ATRA inhibits the expression of CD117^+^ Stro-1^+^ stem cell population in OS models [77]. A case report of a patient with relapsed, refractory OS demonstrated stable complete remission at 14 months after receiving ATRA and interferon-alpha treatment [78]. Further prospective clinical studies of ATRA are needed in OS to determine its clinical utility.

### 5.2. ATS

Plant metabolites, such as terpenoids, have shown anticancer activity in a variety of cancer types because of their anti-inflammatory or immunomodulatory properties [79]. One terpenoid compound, asiaticoside (ATS), which is derived from an herbaceous plant named *Centella asiatica,* has been studied against OS using in vitro and in vivo models. M0 macrophages were stimulated with IL-4/IL-13 to make M2 macrophages. Flow cytometry and western blot techniques were used to analyze the expression of CD206^+^, Arg-1, and IL-10 to confirm M2 markers. After treatment of M2 macrophages with ATS, M2 macrophage markers showed a reduction in expression, whereas no changes were observed in M1 macrophage markers such as CD14^+^ and CD86^+^, suggesting ATS can repolarize M2 macrophages to the M1 phenotype. Further confirmation was conducted using co-culture experiments where M2 macrophages were incubated with OS cells and showed an increased viability of OS cells, whereas the reverse was observed after treatment with ATS, and significant mortality of OS cells was seen with reduced invasion tendency. These findings suggest that the treatment of ATS can repolarize the M2 phenotype to M1 [69].

### 5.3. PTT

Photothermal therapy (PTT) is an alternative therapy designed to enhance the delivery of various antigens, vaccines, or drugs through microneedles, which reduces side effects with selective delivery. PTT uses photothermal agents such as carbon, silica, or gold nanomaterials that can absorb light and result in hyperthermia and subsequent cell death in cells that are in direct contact [80]. Pan and colleagues evaluated the use of polyacrylic-coated gold nanorods as a PTT against OS cells, demonstrating cytotoxicity, but no inhibitory effect was seen on the invasion tendency of the OS cells when observed through in vitro analysis [81]. In a recent study, M2 macrophages were co-cultured with graphene oxide (GO), and the post-treatment administration of PTT resulted in the reduced expression of the M2 marker, CD206^+^ marker, compared to the non-PTT treatment group, suggesting that GO is able to repolarize macrophages as well as show a reduction in the invasive potential of OS cells. Further, an in vivo mice model also suggested reduced tumor burden after the treatment of GO and PTT compared to the control group. Overall, these results suggest the reduced invasive behavior of OS cells after PTT and GO treatment through the induction of pro-inflammatory cytokines [70].

### 5.4. Mifamurtide

Another interesting approach is a synthetic analog of the bacterial cell wall, mifamurtide, which has been used as a treatment regimen against primary OS tumors and has shown to stimulate the anti-inflammatory immune response [71]. In vitro studies using human macrophages suggest a role for mifamurtide in suppressing the progression of OS. Pretreated macrophages with mifamurtide showed the reduced expression of osteoblast markers and resulted in repolarizing the M2 macrophages to M1 through the inhibition of pAKT and STAT3. Authors tried to target the OPG, RANK/RANKL pathway in this study and saw a decreased expression of RANK in mifamurtide-treated macrophages. Therefore, this study suggested the use of mifamurtide in cooperation with the RANK/RANKL pathway antibody for better future treatment strategies against OS [71,82,83].

Mifamurtide has been approved in Europe in combination with standard chemotherapy. In a small phase I study in patients with OS, mifamurtide was seen to be well tolerated [84]. A phase II clinical trial compared 24 versus 36 total doses, demonstrating improved relapse-free survival in those receiving 36 doses compared to those receiving 24, with good tolerability in both groups [85]. A phase III clinical trial was conducted using a combination of MAP with mifamurtide and ifosfamide. No significant impact on EFS was observed with mifamurtide treatment, and the addition of mifamurtide to conventional chemotherapy did not significantly improve clinical outcomes. A recent clinical study did suggest improved outcomes of the combination of mifamurtide and conventional therapy in younger patients with localized OS [86].

### 5.5. Pexidartinib

Colony-stimulating factor-1 (CSF1) has been known to play a role in the differentiation and accumulation of macrophages at the tumor niche and has been known to stimulate the M2-type phenotype in the TME. Pexidartinib, an inhibitor of CSF1, has been used to treat tenosynovial giant cell tumors and underwent clinical trials and showed promising results [87]. Recently, pexidartinib, an inhibitor of CSF1, has repolarized M2 macrophages to the M1 phenotype in both in vitro and in vivo mouse models [72]. A recent phase I multicentric clinical trial was performed in adult patients with advanced soft tissue sarcoma (NCT02584647) who received 600 and 1000 mg of pexidartinib in combination with sirolimus. Tumor samples were stained for CD68^+^, CD163^+^, and CD206^+^, and CSF1R expression was used to determine the phenotypic expression of cells present in the tumor. Clinical benefit was observed in 12/18 patients (66.7%; 95% CI, 41.15–85.64%), with the median PFS found to be 11.6 weeks (95% CI, 6–24.57), and overall survival was 35.9 weeks (95% CI, noncalculable). This combined therapy also reduced the expression of the M2 macrophage CD206^+^ marker in the patient’s tumor tissue, which was during treatment, compared to when observed from pre-treatment tissues from the same patient. Overall toxicity was acceptable, with 28% of patients experiencing elevated levels of aminotransferase and alanine aminotransferase [88].

### 5.6. Resiquimod

Another important repolarization target includes toll-like receptors (TLRs). A study used a toll-like receptor 7/8 (TLR) agonist named resiquimod (R848) in treatment against OS and loaded them with poly-L-histamine bound nanoparticles (NP_R848_). They were further incorporated with cisplatin and linked with hyaluronic acid (HA) (^CDDP^NP_R848_) for targeted drug delivery. Mouse bone marrow-derived macrophages were used to induce M1 and M2 macrophages, and CD86 and CD206 markers were used to identify M1 and M2 stimulation, respectively. M2 macrophages incubated with PBS showed 83% expression of CD206. After treatment with either R848 or NP_R848_, a significant reduction in M2 macrophage expression to 56.6% and 65.8% was observed, respectively, which suggested that it helps re-educate the macrophages to convert into M1 phenotype. The tumor inhibition rate (TIR) was also calculated in mice OS models with different NP treatments, and ^CDDP^NP_R848_ showed the highest TIR rate of 80.5 ± 7.6%, with minimum toxicity to the organs compared to other treatments used. TAM re-education was observed on isolated tumors of mice and on tumor-draining lymph nodes. The M2 macrophage marker showed 8.53% positivity in the control group (PBS), and after treatment with ^CDDP^NP_R848_, the M2 macrophages showed a percentage of 0.48%. These targeted NPs induced an anti-tumor effect with a long-term immune memory effect through DC maturation and TAM repolarization [73].

### 5.7. Esculetin

Coumarin (1, 2-benzopyrone) is another known natural anticancer compound known to be present in cherry plants, and its derivatives such as methoxyl or alkoxyl furocoumarins have shown anticancer properties when used in combination with ultraviolet A [89,90]. Treatment with dihydroxycoumarins (esculetin, fraxetin, and daphnetin) was studied on OS in vitro and in vivo models. Only esculetin showed the inhibition of OS cells in a dose- and time-dependent manner through cell cycle arrest at the S phase of the cells. THP1 cells were used to produce M1 and M2 macrophages, and esculetin reduced the production of M2 anti-inflammatory macrophage cytokines such as IL-10 and TGF-beta, whereas no effect was observed on M1 macrophages. Daphnetin did not show any mortality effect on OS cells or any significant changes in macrophage phenotype. Esculetin and fraxetin showed mouse tumor reduction as well as reduction in lung metastasis. Additionally, liver metastasis was observed to be inhibited by the treatment of esculetin and fraxetin [74].

## 6. Other Strategies to Target TAM/M2 Macrophages

Macrophages present in the TME have also been found to be targeted by other strategies such as the enhancement of phagocytic activity, and enhanced antigen presentations in respect to OS are mentioned below.

### 6.1. CD47/SIRPα

OS TME is known to be infiltrated with TAM, which masks them from phagocytic or killing effects of macrophages by expressing the CD47 marker on the surface, which result in the non-killing of these tumor cells. CD47 is a transmembrane protein that acts as a ligand for signal regulatory protein-α (SIRPα), which is expressed on macrophages, and this interaction inhibits the phagocytic property of macrophages, which leads to increased tumor growth via less antigen presentation [91,92]. A previous study by Cao et al. assessed the genetically engineered oncolytic vaccinia virus (VV) to express SIRPα-Fc (SIRPα-Fc-VV), which helps macrophages to secret this SIRPα and redirect the phagocytic property to control the tumor cell proliferation [93].

A previously published study suggested that the utilization of CD47mAb therapy in OS cells enhances the phagocytic property of ferumoxytol iron-nanoparticles. The co-culture of bone marrow-derived macrophages with OS cells and treatment with CD47 antibodies suggested the three to five-fold increased phagocytosis of OS cells in vitro. The enhanced uptake of iron nanoparticles was also observed in the primary bone tumor of mice using MRI and Pearl’s staining [94].

CD47 antibody has also been studied in combination with the standard chemotherapy drug doxorubicin to enhance anticancer activity [95]. Doxorubicin is known as a triggering agent for calreticulin, which provides the signal of “eat me” against TAM. In this study, bone marrow-derived M1 macrophages were co-cultured with OS cells and treated with CD47mAb alone, with doxorubicin alone, or in combination. The increased phagocytic property of M1 macrophages was found when treatment was given with combination therapy compared to either of the monotherapy used. The in vivo studies using this combination therapy enhanced antitumor activity by reducing the intratibial tumor growth and suppressing pulmonary metastases. The combination therapy reduced the primary tumor flux in mice by seven-fold compared to doxorubicin alone and four-fold less when observed with CD47mAb. Lung histopathology suggested no significant change in tumors in control mice when compared to doxorubicin-alone-treated mice [95]. Another recent study suggested the high expression of CD47 expression along with M1 macrophage marker expression and suggested the role of the IL-18 cytokine, which is released by M1 macrophages, leading to enhanced CD47 expression. The increased IL-18 expression was further correlated with the increased production of L-amino acid transporter 2 (LAC-2) and was suggested to be a new target for treatment in OS [96].

### 6.2. CD40

Another potential target molecule in cancer immunology is CD40, which is a TNF Receptor Superfamily (TNF-R-SF) signaling molecule required for antigen presentation and is an effective immune response. A previous study conducted by Zhang et al. used CD40 ab alone and checkpoint inhibitor antibodies, such as PD-1 and CTLA-4, alone and in combination and suggested that this combination resulted in high CD8^+^/Treg cells and the lower expression of PD-1 on T cells. When the anti-PD-1 antibody was combined with CD40 ab, it resulted in decreased tumor burden in mice when compared to the single antibody treatment [97]. Authors also found that the single treatment with CD40 ab therapy in mice decreased the PD-1 expression on tumor cells and also changed the T cell phenotype, which helped increase the expressions of IFN-gamma, Ki-67, and granzyme-B, helping clear the tumor [97].

### 6.3. SIGLEC-15

Sialic acid-binding immunoglobulin-like lectins (Siglecs) are a family of membrane proteins that display an amino-terminal domain that binds to sialic acid and has similarities to immunoglobulin domains. These proteins are subdivided into two subgroups. The first group is CD33, which is present on myeloid linage cells, and another sub group contains- siglecs-1, siglecs-2, siglecs-4, and siglec-15. Among siglecs, siglec-15 has been found to play a role in bone growth and osteoclast differentiation, and it was also reported that this signaling molecule can be responsible for giant cell tumors of bones [98,99]. A recent study demonstrated the expression of siglec-15 on OS cells using various cell lines and concluded that this gene promotes the proliferation of OS cells along with increased invasion tendency and migration properties in vitro [100]. When gene silencing for siglec-15 was performed in various OS cells, they were unable to proliferate in the mouse model, and it resulted in a smaller tumor size when compared to the control cells with normal siglec-15 expression. Dual specificity protein phosphatase 1 (DUSP1) was used as a tumor suppressor and also affected the MAPK pathway, and it has been known to be expressed in various myeloid cells [101]. DUSP1 expression was found to be upregulated in OS cells when siglec-15 silencing was performed, which suggested that MAPK/DUSP1/Siglec-5 are potential therapeutic targets for OS as the former compensates the effect of the other and leads to the proliferation of OS cells in in vitro and in vivo models [100].

### 6.4. Complement

Complement system activation has been shown to regulate innate and adaptive immune responses and other inflammatory reactions [102]. It is known that two essential complement activators, C3a and C5a, help to modulate the osteoclast formation and balance the formation of osteoblast activation in the bone TME, enhancing the progression of tumor growth [103]. These two complement activators also regulate macrophage-induced angiogenesis but show opposing effects if attached to their counter receptors, C3aR and C5aR, respectively [104]. A study conducted by Jeon et al. showed that OS cells show an increased complement activation pathway (alternative pathway) when induced under the influence of growth factors and showed increased angiogenic activity in endothelial cells. The increased production of VEGF-A and FGF-1 was seen in OS cells, which showed complement activation [105]. Another recent study by Takemoto et al. suggested the use of antibody PG4D2 and A201, which target Podoplanin (PDPN), which is a transmembrane protein expressed in multiple tissues [106]. High expression of PDPN on OS tumor cells has been correlated with poor prognosis and has shown to be dominantly expressed in metastasis over primary tumors [107]. PDPN-expressing tumor cells activate the ability of platelets to activate the C-type lectin-like receptor 2 (CLEC-2). The interaction between the PDPN-CLEC-2 increases the angiogenesis and progression of tumor and metastasis. The use of PG4D2 antibodies has reduced tumor growth in xenograft OS mouse models. These antibodies have suggested antibody-independent but complement-dependent lysis through the inhibition of the interaction between the PDPN-CLEC-2, which showed the progression of OS [108].

### 6.5. PI3K

Class 1 phosphoinositide -3 kinase (PI3K) is a regulatory enzyme that is known to phosphorylate inositol ring at the 3′ position and generates a phosphatidylinositol-3,4,5-trisphosphate, a secondary messenger responsible for the generation of various signaling pathways at the plasma membrane of the cells [109]. Class I PI3K is further divided into class 1A, which consists of PI3Kα, PI3Kβ, and PI3Kδ, which are activated by receptor tyrosine kinase (RTKs), and class 1B consists of PI3Kγ, which is associated with G-protein coupled signaling [110,111]. Previous studies have suggested that PI3Kγ signaling leads to tumor progression and promotes M2-type macrophage activation in the TME by blocking CD8^+^ T cell cytotoxic effect on tumor cells [110,111,112]. Previously, the role of PI3Kγ was studied using KO mice and showed a reduction in tumor progression in lung and breast carcinoma, with an enhanced pro-inflammatory response and prolonged survival of mice. As PI3Kγ signaling is known to be present only in macrophages and not in tumor cells, its inhibition could switch the anti-inflammatory response to the pro-inflammatory response. These findings were further concluded when mRNA and the protein expression of murine macrophages lacking PI3Kγ signaling showed increased pro-inflammatory cytokine response. This study highlights the role of PI3Kγ signaling as a targeted therapy for macrophage repolarization [111]. Various PI3K inhibitors have been studied in the past against OS [113]. Alantolactone, which is a natural molecule isolated from *Inula helenium*, has previously been shown to inhibit the proliferation of OS cells through the inhibition of the PI3K signaling pathway, but the detailed mechanism of action in vivo is still unexplored [114]. On the basis of the PI3Kγ pathway and its known mechanism, it could be a potent target against OS.

### 6.6. ERK5

Extracellular signal-regulated kinase 5 (ERK5) is a member of the mitogen activated protein kinase (MAPK) family, which is activated in response to several growth factors or stress [115]. ERK5 has been associated with cell cycle regulation, and MAPK-ERK5 signaling has suggested its role in cancer progression [116]. Previous studies using ERK5 KO mice suggested that the mice macrophages deficient in ERK5 in myeloid lineage increased the M1 phenotype expression, whereas low M2 phenotypic expression was found in the bone marrow-derived macrophages of the mice. Furthermore, these results were confirmed by analyzing cell surface markers in a xenograft model, which showed high expression of M1 cytokines, such as iNOS and IL-12beta; low expression of Arg1, TGF-beta, and IL-10; and impaired STAT3 signaling [117]. The majority of TAM population expression was further confirmed using melanoma graft mice models, in which a high number of KI67^+^ and CD163^+^ cells were present in mice with ERK5 expression, and an approximately 50% reduction in the expression of these cell surface markers was observed in the KO mice [118]. These results suggested that ERK5 regulates TAM-type signaling in the TME, and it helps in the differentiation of macrophages. Another study analyzing patient and mice tumors using RNA sequencing collectively support that MAPK7/MMP9 (matrix metallopeptidase 9) is responsible for inducing the primary bone cancer progression resulting in tumor/metastasis growth, colony formation, and macrophage polarization [119]. Therefore, these studies suggest that MAPK or ERK5 signaling could be a targeted strategy to induce repolarization in OS.

## 7. Conclusions

OS treatment strategies have developed over the decades and improved with the introduction of immunotherapy, but they still are not very effective and do not provide a full response in patients. Many factors are accountable for the failure of immunotherapy in OS, which includes TME or the presence of TAM in the tumor or metastatic tissue and the infiltration of macrophages, immune cells, and other bone cells. OS is a form of solid tumor, and therefore, it becomes hard to permeate the thick fibrous primary tissue, which further impedes the immunotherapy. Other factors may also include resistance to therapy and lack of contact between T cells and tumor cells for effective stimulation, which halts insufficient IFN-gamma signaling for T cells to secrete or stimulate cytokines responsible for suppressing the tumor. Other important factors include a lack of immune tumor cells or antigen presentation, which cannot provide enough antigen to present on the surface of tumor cells to be recognized by T cells. Some studies have suggested that immunotherapy can cause liver or kidney toxicity or even cardiac arrest to patients.

As the success of immunotherapy is not very effective, there is an utmost need for alternative strategies to target OS, such as primary tumor and secondary lung metastasis, which can be performed by repolarizing the TAM/M2 macrophages to M1 macrophages to induce the cytokine and activation of antitumor effect. Therefore, in this review, we listed and discussed recent studies that have used drugs or natural immune agents to repolarize TAM/M2 macrophages to M1. Therefore, targeting a specific cell population in the TME niche instead of having the hurdle of choosing many cell types as a target (in the case of checkpoint inhibitors) can overcome the problem of tumor progression. Therefore, reshaping or re-educating the target macrophage population is the holy grail of therapeutic macrophage targeting to suppress tumor growth and to achieve tumor inhibition.

### Future Perspectives

In the present manuscript, we discussed the previously known role of macrophage polarization and its interaction with the bone microenvironment and their influence on targeting OS in its progression or inhibition. We summarized how macrophages change their polarization in TME in response to external stimulus and behave as a friend or foe using various in vitro and in vivo models of OS. We found that the repolarization of M2/TAM macrophages is a hot area for cancer immunotherapy because it gives a broader area of focus with respect to targeted therapy against various types of cells involved in TME. Therefore, the repolarization of M2/TAM should be studied in more detail because of its diverse nature.

## Figures and Tables

**Figure 1 ijms-24-02858-f001:**
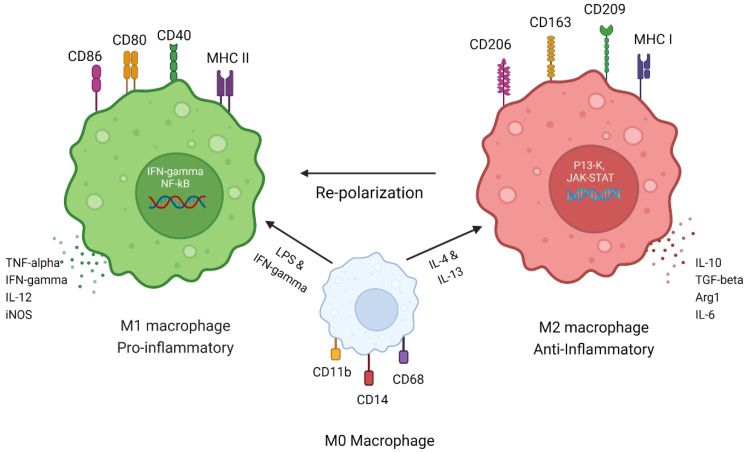
Macrophage plasticity. Macrophage states under different stimuli. Monocyte-derived M0 macrophages have CD11b, CD14, and CD68 as their surface markers. The stimulation of M0 macrophages with LPS and IFN-gamma converts them to pro-inflammatory M1 macrophages, which express cell surface receptors such as CD86, CD80, CD40, and MHC class II. M1 macrophage secretes pro-inflammatory cytokines such as TNF-alpha, IFN-gamma, IL-12, and iNOS. M1 macrophages have IFN-gamma signaling and NF-kB signaling, which stimulates them and preserves the M1 phenotype. M0 converts to M2 macrophages after stimulation with anti-inflammatory cytokines such as IL-4 and IL-13 and expresses cell receptor markers CD206, CD163, CD209, and MHC class I. M2 macrophages secrete anti-inflammatory cytokines such as IL-10, TGF-beta, Arg-1, and IL-6. M2 macrophages function through P13-K and JAK-STAT pathways and maintain their M2/TAM phenotype. Macrophage function varies with different stimuli and can perform anti-tumor (M1) or tumor-supportive (M2) functions. Adapted from “Immune Therapy in Multiple Myeloma”, by BioRender.com (2023). Retrieved from https://app.biorender.com/biorender-templates (accessed on 30 January 2023).

**Figure 2 ijms-24-02858-f002:**
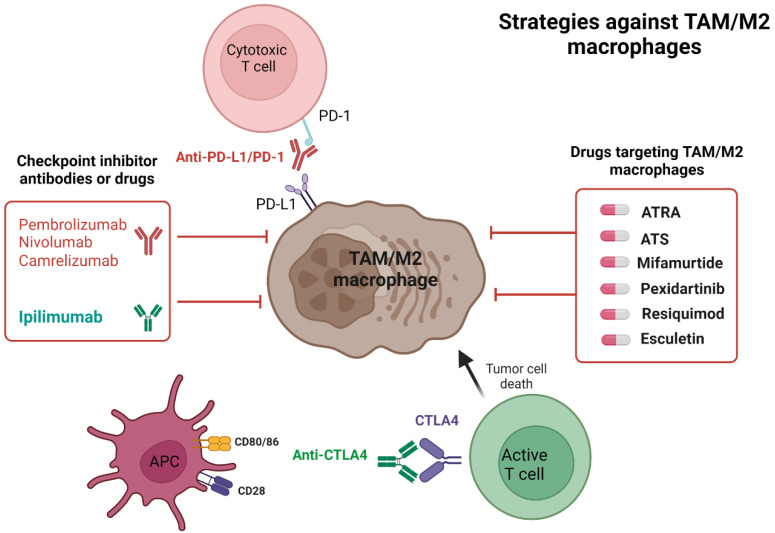
TAM/M2 macrophage targeting strategies. In the tumor microenvironment, T cells express PD-1, which can interact with its ligand PD-L1 on cancer cells as well as M2 macrophages, resulting in the inactivation of T cell function. Anti-PD-L1/PD-1 antibodies block this interaction, and T cells are subsequently activated and have anti-cancer activity. The repolarization of M2 macrophages to the M1 phenotype also reduces PD-L1 expression in the tumor microenvironment. Various agents that can re-polarize TAMs/M2 macrophages are being developed, and those under development as strategies to target the macrophage population are shown in the image. Adapted from “Immune Therapy in Multiple Myeloma”, by BioRender.com (2023). Retrieved from https://app.biorender.com/biorender-templates (accessed on 30 January 2023).

**Table 2 ijms-24-02858-t002:** Drugs targeted against TAM/M2 macrophages using in vitro and in vivo studies.

Drug	Target Cell Type	Markers Used for Flow Cytometry/IHC/RT-PCR	Inhibition In Vitro Drug Concentration	Target Cells Type: In Vitro/ Primary Tumor/Pulmonary Metastasis/In Vivo	Mechanism
All-Trans Retinoic Acid (ATRA) [68]	TAM/M2	F4/80, CD206+ CD209, CD86 CD14	Pretreatment of mice for 7 days at 20 mg/kg and post injection 40 mg/kg for 4 weeks	ATRA reduced TAM macrophage polarization in vitro. Secondary lung macroscopic metastatic reduction was seen to 60% and 95% after 1- and 2-weeks treatment respectively.	MMP12 Inhibition from M2 macrophages to suppress metastasis
Asiaticoside (ATS) [69]	M2	CD206, CD14, CD86, Ki67, Bcl-2, Bax, VEGF	40 µM invitro and 10 mg/kg in vivo every 2nd day for 30 days	ATS restrained the M2 phenotype and helped reduce the tumor weight by 3-fold and suppressed OS progression.	TRAF6/NF-kB inhibition
Graphene Oxide (GO) mediated Photothermal therapy (PTT) [70]	M2	CD206, CD209, Arg-1	0.05 mg/mL in vitro and 808 nm light (0.7 W/cm^2^, 1.5 min in vivo, temperature ≥45 °C	Low-temperature PPT helped polarize to M1 phenotype and show antitumor effects.	Suppression of IL-2 induced M2 repolarization
Mifamurtide [71]	M2	CD11b, CD3, CD45.2, Ly6.G, MMP2/ MMP9, TNF-Ƴ, TRPV1	100 µM in vitro and 5 mg/mL in vivo	Treatment showed a reduction in osteoblast markers. M1 treated cells showed increased iron transporter expression of DMT1.	Inhibition of STAT3 pathway/anti RANKL therapy
Pexidartinib (PLX3397) [72]	TAM/M2	CD206, CD86, iNOS, IL-1beta, CD80, CD206, CCL2	10 mmol/L in vitro In vivo 5 and 10 mg/kg	Treatment showed suppression of TAM phenotype and increased chemotaxis. The mouse model showed suppressed primary tumor and metastasis and possibility of transition into immunotherapy.	Inhibition of CSF1/CSF1R signaling
Resiquimod cisplatin loaded nanoparticle (^CDDP^NP_R848_) [73]	TAM/M2	CD86, CD206 CD44, CD62L	10 µg/mL	Treatment effectively suppressed the tumor growth in vivo and stimulated the induction of immune memory response in spleen.	D88-dependent signaling pathway
Esculetin and fraxetin [74]	M2	Cyclin D1 and CDK4	In vitro- 10–100 µM In vivo- 3 or 10 mg/kg for 35 days	Esculetin showed cell cycle arrest at S phase and differentiation of M2 macrophages. Esculetin and fraxetin showed antitumor activity against primary and secondary metastatic cancer.	Inhibition of M2 macrophage differentiation

## Data Availability

Not applicable.

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
