# Peer review of "Macrophage Repolarization as a Therapeutic Strategy for Osteosarcoma"

_ijms, 2023, doi:10.3390/ijms24032858_

Round 1

Reviewer 1 Report

The authors have produced a review on therapeutic strategies for osteosarcoma based on macrophage repolarization. This review is very clear, complete and well organized with tables and figures to summarize. All data is well referenced.

This review can be published without modification. 

Author Response

Respected Reviewer

Thank you for taking your valuable time and reviewing our manuscript. We highly appreciate your positive response to our manuscript and acceptance of it for publication. 

Reviewer 2 Report

In this manuscript, Namrata Anand and collaborators reviewed recent studies regarding the M2-M1 repolarization strategies in treating OS cancer. Overall, this review is well written and organized.

 Comments/suggestions:

1)     References should be updated: a recent review (PMID: 32717819) focused on the role of TAMs in the complexity of TME in OS should be included.

2)     Some of the molecular targets that are involved in TAMs repolarization are missing: For example,Kaneda et al., (PMID: 27642729) demonstrated that PI3Kγ controls the TAM switch between immune suppression (M2) and immune stimulation (M1). Giurisato et al., (PMID: 29507229)  identified ERK5 as a pro-tumor macrophages that is involved in TAMs repolarization and in M2-like TAM self-renewal (PMID: 32561530). These molecular targets should be discussed and considered as a potential therapeutic strategies in the context of OS treatment.

Author Response

Reviewer #2 Comments:

Point 1)  References should be updated: a recent review (PMID: 32717819) focused on the role of TAMs in the complexity of TME in OS should be included.

Author’s Response: Respected Reviewer, thank you for taking your valuable time and reviewing the manuscript, we highly appreciate it. We apologize for not mentioning the publication on TAM in OS before, but now as per your suggestions, we have incorporated the citation in our manuscript at page number 5, changes can be seen in track changes (red in color).

Point 2)  Some of the molecular targets that are involved in TAMs repolarization are missing: For example,Kaneda et al., (PMID: 27642729) demonstrated that PI3Kγ controls the TAM switch between immune suppression (M2) and immune stimulation (M1). Giurisato et al., (PMID: 29507229)  identified ERK5 as a pro-tumor macrophages that is involved in TAMs repolarization and in M2-like TAM self-renewal (PMID: 32561530). These molecular targets should be discussed and considered as a potential therapeutic strategies in the context of OS treatment.

Author’s Response: Thank you for taking your time and pointing out that we have missed some of the important studies. With the reviewer’s suggestion, we have incorporated the above-mentioned studies on page number 22 and 23 in track changes and their role has been mentioned as described as a targeted strategy for tumor macrophage repolarization and has been mentioned with the proper citation in the manuscript.